# CD44-Receptor Targeted Gold-Doxorubicin Nanocomposite for Pulsatile Chemo-Photothermal Therapy of Triple-Negative Breast Cancer Cells

**DOI:** 10.3390/pharmaceutics14122734

**Published:** 2022-12-06

**Authors:** Dnyaneshwar Kalyane, Suryanarayana Polaka, Nupur Vasdev, Rakesh Kumar Tekade

**Affiliations:** National Institute of Pharmaceutical Education and Research (NIPER), Ahmedabad, An Institute of National Importance, Opposite Air Force Station, Department of Pharmaceuticals, Ministry of Chemicals and Fertilizers, Government of India, Gandhinagar 382355, India

**Keywords:** gold nanocomposite, chemo-photothermal therapy, breast cancer, CD44 receptor targeting, ROS generation, thermal ablation

## Abstract

This study reports the CD44 receptor-targeted gold-doxorubicin nanocomposite (*_T_*GNC-DOX) for pulsatile chemo-photothermal therapy of triple-negative breast cancer (TNBC). The developed *_T_*GNC-DOX was nanometric, having a particle size of 71.34 ± 3.66 nm. The doxorubicin was loaded by electrostatic interaction with high entrapment and loading efficiency (>75%). *_T_*GNC-DOX showed potent photothermal response and reversible photothermal stability following irradiation with 808 nm NIR laser irradiation. Further, *_T_*GNC-DOX showed laser-responsive and pH-dependent drug release behavior suggesting its suitability for chemo-photothermal therapy, specifically at the tumor microenvironment site. Cellular viability, cellular uptake, ROS generation, and apoptosis assays suggested selective localization of *_T_*GNC-DOX in cancer cells that showed a significant cytotoxic effect against MDA-MB-231 breast cancer cells. Moreover, the developed *_T_*GNC-DOX showed ferroptosis in MDA-MB-231 cells. The event of *_T_*GNC-DOX-mediated thermal ablation is marked by a significant generation of reactive oxygen species (ROS) and apoptosis, as affirmed by flow cytometry. NIR-808 laser-responsive photothermal therapy of cancer cells was found to be more effective than without NIR-808 laser-treated cells, suggesting the fundamental role of photothermal ablation. The outcome concludes developed *_T_*GNC-DOX is a novel and potential tool to mediate laser-guided chemo-photothermal ablation treatment of cancer cells.

## 1. Introduction

Solid tumors have become a huge concern throughout the world. This problem urgently needs to be addressed and efficient cancer therapies should be employed to completely eradiate tumors. Among solid tumors, triple-negative breast cancer (TNBC) is one of the prominent causes of the death in women [1]. Photothermal therapy (PTT), a hyperthermia based approach, recently emerged as alternative therapy to cure the cancer. This therapy generates PTT-based hyperthermia in localized tumors in the presence of a photothermal agent or photosensitizer upon near infrared (NIR) irradiation [2,3,4]. Moreover, PTT minimizes damage to the non-targeted tissues by selectively targeting to the tumor tissues [5]. Despite selectivity, PTT alone is not efficient to completely eradiate the tumors and there are chances of some of the residual cancer cells may remain at the tumor site, which further leads to cancer reoccurrence and tumor progression [6,7]. Furthermore, this therapy is not practically possible with disseminated, metastatic tumors, which are inaccessible to NIR light. To resolve this issue, combination strategies have been employed to improve therapeutic efficacy. Chemo-PTT is one of the emerged combination strategy approaches used to complete the removal of the tumor [2,7,8].

Gold nanoparticles (GNCs) are biocompatible photothermal agents and are widely employed in PTT due to their strong surface plasmon resonance [9,10,11,12]. These nanoparticles show strong NIR absorption and great capability of light to heat conversion [13,14].

Doxorubicin (DOX) has been extensively used to treat breast cancer. It exerts anti-tumor action by interaction with DNA replication [15]. However, it shows systemic toxicity and many side effects; hence it needs to be delivered at the tumor site. The synthesizing GNCs below 100 nm will promote their selective accumulation at the tumor site through the enhanced permeability and retention (EPR) effect [16]. The loading of DOX in GNCs will minimize the side effects and deliver the drug to tumor tissue [17,18]. Moreover, after NIR irradiation, an effective photothermal effect will be achieved with enhanced DOX release [19].

It is reported that CD44-receptors become overexpressed in TNBC cells [20]. Targeting these receptors can be achieved by conjugating hyaluronic acid (HA) to the GNCs, further enhancing cellular uptake of GNCs and improving cancer therapy [21].

In this study, we employed previously synthesized *_T_*GNCs to synthesize *_T_*GNCs DOX. These *_T_*GNCs were synthesized by one pot strategy in which HA used as a reducing agent and targeting ligand for CD44 receptor. The synthesized *_T_*GNCs DOX were investigated for photothermal efficiency and reversible photothermal effect which is essential to achieve good photothermal effect. Moreover, the drug release studies were performed at pH 7.4 (blood pH) and 6.5 (tumor pH) as well as effect of NIR irradiation on drug release was studied. A cellular blockade assay was performed to investigate receptor-mediated endocytosis/cellular uptake of *_T_*GNC-DOX in the presence and absence of HA and anti-CD44 antibody. Moreover, intracellular reactive oxygen species (ROS) generation, apoptosis assay, and cell cycle analysis were performed to check the ROS generation property of *_T_*GNC-DOX, apoptosis, and cell cycle arrest by *_T_*GNC-DOX, respectively. The Ferroptosis effect of *_T_*GNC-DOX in MDA-MB-231 cancer cells was evaluated using a BODIPY C11 probe through confocal laser scanning microscopy and flow cytometry.

## 2. Result and Discussion

### 2.1. Particle Size, Polydispersity Index, and Surface Charge Determination

The particle size of optimized *_T_*GNC-DOX was 71.34 ± 3.66 nm with PDI 0.43 ± 0.02 as determined by DLS using the Malvern zeta sizer via the NIBS 173° backscattering technique (Figure 1A). We have considered the particle size of *_T_*GNC-DOX to be less than 100 nm to attain better-enhanced permeability and retention effect. The surface zeta potential of *_T_*GNC-DOX was found to be −13.86 ± 0.61 mV, which was increased from −21.80 ± 0.17 mV before DOX loading, indicating electrostatic interaction between positively charged DOX and negatively charged HA (Figure 1B).

### 2.2. Surface Morphology of Synthesized _T_GNC-DOX

FE-SEM used to investigate the structure and surface morphology of *_T_*GNC-DOX. FE-SEM images of *_T_*GNC-DOX revealed that the developed *_T_*GNC-DOX shows uniform particle size (Figure 1C,D). The particle size by FE-SEM analysis was found to be 46.73 ± 5.11 nm (*n* = 50).

### 2.3. Topography Attributes of _T_GNC-DOX

TEM is essentially used to characterize nanocomposites due to its high-resolution properties. This imaging technique helps to get accurate size, shape, structure, and composition of nanocomposites. For that, to get the proper structure and composition of *_T_*GNC-DOX, TEM imaging was employed. In this study, HA templated *_T_*GNC-DOX showed a coating of HA, which can act as a target for the CD44-receptor and assist in loading DOX (Figure 2A,B). The particle size by HR-TEM analysis was found to be 43.30 ± 5.27 nm (*n* = 50).

### 2.4. UV Absorption Profile

UV spectroscopy was performed to observe the characteristics absorption peak of *_T_*GNC-DOX. The UV spectrum showed high absorbance in the NIR region, indicating the NIR laser absorption capability of the developed *_T_*GNC-DOX (Figure 2C).

### 2.5. Fourier-Transform Infrared (FTIR) Analysis

The FTIR study was performed to evaluate the functional groups present on *_T_*GNC-DOX and the successfully loading of doxorubicin in *_T_*GNC-DOX. The FTIR peaks at 1577 cm^−1^ and 1611 cm^−1^ confirm C=C ring stretching; peaks at 797 cm^−1^ and 692 cm^−1^ depict the C=H bend and C=C ring bend, respectively. This confirms the doxorubicin has been successfully loaded in *_T_*GNC-DOX (Figure 3) [22]. Moreover, other peaks at 3402 cm^−1^ for the hydroxyl group, and 2922 cm^−1^ for C-H stretching association of peaks of both HA and doxorubicin. The peak at 1727 cm^−1^ for C=O stretch shows association of peaks with DOX.

### 2.6. Confirmation of HA Surface Functionalization

It is well reported that HA act as a ligand for CD44 receptor, which becomes overexpressed in cancer cells. Hence, to confirm the HA in developed *_T_*GNC-DOX recombinant CD44 protein was used, which shows strong affinity towards HA. The obtained results revealed that *_T_*GNC-DOX showed higher green fluorescence compared to Tetrakis-reduced conventional gold nanoparticles which is shown in Figure 4. This confirms presence of HA in developed *_T_*GNC-DOX.

### 2.7. X-ray Diffraction Analysis

The XRD investigates the amorphous or crystalline nature of any substances. The XRD pattern of *_T_*GNC-DOX showed four identical Bragg’s reflection peaks at 2 θ values, i.e., 38.32 (111), 44.51 (200), 64.79 (220), and 77.69 (311), reflecting the face-centered cubic structure of gold as reported in earlier reports (Figure 5) [23,24]. While DOX showed intense diffraction peaks at different 2 θ values, after loading into *_T_*GNC-DOX, the intense crystalline peaks of DOX were not observed, indicating that DOX was successfully loaded in *_T_*GNC-DOX [25,26].

### 2.8. Photothermal Behavior

To evaluate the photothermal response, *_T_*GNC-DOX dispersion in Milli-Q water (20 µg/mL) was continuously irradiated under 808 nm NIR laser at a laser power of 2.4 W/cm^2^ for 15 min. The temperature changes concerning the irradiation time were monitored using a thermal infrared camera (Optis PI, Berlin, Germany). The temperature was noted to be increased to 50.56 ± 0.41 °C after 15 min NIR irradiation (Figure 6A).

### 2.9. Photothermal Stability of _T_GNC-DOX

To investigate photothermal stability and reversible photothermal effect, *_T_*GNC-DOX was continuously irradiated under 808 nm NIR laser at a laser power of 2.4 W/cm^2^ for 15 min, followed by cooling for 15 min (30 min cycle). After repeated heating and cooling cycles, the developed *_T_*GNC-DOX did not show a change in final temperature, which indicates the good photothermal potential of *_T_*GNC-DOX (*p* > 0.05) (Figure 6B).

### 2.10. Determination of Drug Loading and Entrapment Efficiency

The efficient drug loading and entrapment in nanomaterials is highly dependent on the interaction of nanomaterials with the drug molecule. In this study, developed *_T_*GNC-DOX showed high DOX loading and entrapment efficiency, and it was found to be 78.97 ± 0.35% and 75.13 ± 1.61%, respectively (Figure 7).

The high DOX loading and entrapment was the effect of electrostatic interactions between positively charged DOX molecules and negatively charged HA layer covered on the *_T_*GNC-DOX [27].

### 2.11. In Vitro Drug Release and Stability Studies

In vitro drug release study results revealed that 22.11 ± 1.40% DOX was released at pH 7.4; the release rate increased to 27.86 ± 1.08% at pH 6.5 after 72 h (Figure 8). These results suggest that DOX shows a pH-dependent release profile and the amount of DOX released was higher under tumor microenvironment conditions (pH 6.5). This may be due to decreased electrostatic interactions between negatively charged HA and positively charged DOX at tumor microenvironment acidic pH (pH, 6.5) [28].

Upon 808 nm NIR laser irradiation, the release rate of DOX was found to be further increased. The percentage of DOX released after 72 h was significantly higher at pH 6.5 compared to pH 7.4. The percentage of DOX released after 72 h was found to be 33.42 ± 0.82 and 42.21 ± 2.19 at pH 7.4 and 6.5, respectively (Figure 8). The increase in DOX release from *_T_*GNC-DOX after NIR laser irradiation may be due to the temperature effect, which causes a decrease in electrostatic interaction between HA and DOX.

The stability studies revealed that *_T_*GNC-DOX showed particle size and PDI of 74.85 ± 2.24 and 0.42 ± 0.015 respectively after 15 days while zeta potential measurement showed value of −14.36 ± 0.55. This indicates *_T_*GNC-DOX shows both physical and chemical stability.

### 2.12. Cellular Uptake Potential

*_T_*GNC-DOX showed significant cellular uptake and accumulation as observed under a confocal microscope and quantified using ICP AES. On the other hand, cellular uptakes under the influence of free HA treatment (10 mg/mL) before *_T_*GNC-DOX have shown a reduction in its uptake into the cancer cells (Figure 9A,B).

This suggests that HA pretreatment blocks the CD44 receptors on the cell’s surface and, thus, competes with the receptor-mediated endocytosis of *_T_*GNC-DOX [29]. Moreover, the receptor mediated endocytosis further investigated using anti-CD44 antibody, the results revealed blocking of CD44 receptors on cell surface by anti-CD44 antibody resulted in decrease in DOX fluorescence intensity compared to without anti-CD44 antibody treatment (Figure 9A,C). Notably, similar results were obtained when samples were analyzed using ICP-AES. Under normal treatment conditions, cellular uptake of *_T_*GNC-DOX was found to be higher, i.e., 5.07 ± 0.25 µg intracellular gold concentration, compared to pretreatment with HA, i.e., 4.72 ± 0.16 µg after 6 h (*p* < 0.05).

Furthermore, CSLM results revealed that *_T_*GNC-DOX, after entering into cells, causes the release of DOX (Figure 10).

### 2.13. Effect of Laser-Directed Ablation on Cytotoxicity of _T_GNC-DOX in MDA-MB-231 Cells

The *IC*_50_ value for DOX was determined by varying the concentration range from 0.1 µM to 1000 µM, and *IC*_50_ value was found to be 2.42 µM. This concentration of DOX was considered for other cell-based experiments. The photothermal cytotoxicity of *_T_*GNC-DOX was evaluated using Alamar blue assay. The MDA-MB-231 cells showed significant photothermal cytotoxicity after *_T_*GNC-DOX treatment, followed by 808 nm NIR irradiation at laser power of 2.4 W/cm^2^ for 15 min (Figure 11). The cell viability of *_T_*GNC-DOX treatment followed by 808 nm NIR irradiation group was 6.20 ± 0.59%. In contrast, cells without NIR laser irradiation showed 53.64 ± 4.05% cytotoxicity (*p* < 0.0001).

### 2.14. Effect of _T_GNC-DOX-Directed Laser Ablation on Intracellular ROS Generation

It is reported that PTT promotes excessive ROS generation, leading to cancer cell death [30,31]. Our study found that *_T_*GNC-DOX significantly promotes ROS generation upon 808 NIR laser irradiation in MDA-MB-231 cells (Figure 12). The *_T_*GNC-DOX treatment showed 68.05 ± 0.60% ROS generation, while upon NIR irradiation, the ROS generation was significantly increased to 83.04 ± 2.95% (*p* < 0.001). The DOX is also known for inducing ROS generation mediated cell death in cancer cells [32], which showed 76.03 ± 0.76% of total ROS generation.

### 2.15. Effect of _T_GNC-DOX-Directed Laser Ablation on Apoptosis in MDA-MB-231 Cells

Apoptosis assay is performed to detect the number of cells undergoing apoptosis followed by treatment with a drug or nanocomposites. In this study, the apoptosis in MDA-MB-231 cells treated with *_T_*GNC-DOX was investigated by flow cytometric analysis of annexin V-FITC/PI double-stained cells. It was found that the laser-irradiated group showed 86.02 ± 0.40% cell apoptosis, which was significantly higher than the non-irradiation group, i.e., 74.58 ± 3.20% (*p* < 0.001) (Figure 13). The DOX-treated group showed 80.15 ± 0.71% of total apoptosis. This suggests *_T_*GNC-DOX shows good anticancer potential mediating chemo-PTT-directed cell apoptosis.

### 2.16. Effect of _T_GNC-DOX-Directed Laser Ablation on Cell Cycle

The cell cycle analysis was performed using a flow cytometer to investigate cell cycle arrest after different treatments. Figure 14 shows the cell cycle analysis of different treatment groups. From the obtained results, it can be concluded that DOX shows G2/M phase cell cycle arrest as reported in earlier studies which account for 44.70 ± 4.71% of the cell population [33]. While the treated *_T_*GNC-DOX group showed predominantly cell cycle arrest in G0/G1 phase, i.e., 69.35 ± 3.24% cell population compared to 51.18 ± 0.53, 36.90 ± 0.95, 63.12 ± 1.34% of control, DOX, and *_T_*GNC-DOX alone group, respectively (*p* < 0.001, *p* < 0.05).

### 2.17. Lipid ROS Detection

Lipid peroxidation indicates ferroptosis, in which highly lipid peroxides are generated, and cells cannot remove these toxic peroxides, leading to cell death [34]. Apart from apoptosis, recent studies report that DOX increases the labile iron pool in cells which is toxic, leading to ferroptosis [35]. Moreover, another recent study investigates the role of GNC in ferroptosis after NIR laser irradiation. From this study, Valle et al. reported that platinum decorated-gold nanostars delivered PTT/ferroptosis therapy in multidrug-resistant cancer cells [36]. Hence, to determine the effect of developed *_T_*GNC-DOX on lipid ROS generation, we performed a lipid ROS generation assay using BODIPY C11.

The BODIPY C11, which is named a lipid peroxidation sensor, detects lipid ROS and causes a shift of fluorescence intensity from 590 nm (red) to 510 nm (green) in the presence of lipid ROS due to oxidation of the probe [37]. Our study found that *_T_*GNC-DOX after NIR laser irradiation showed higher green fluorescence intensity compared to control, DOX alone, and *_T_*GNC-DOX without NIR laser treatment groups. This indicates that *_T_*GNC-DOX after NIR laser treatment causes significant lipid peroxidation and can lead to ferroptosis-induced cell death (Figure 15A).

Moreover, flow cytometry analysis also revealed that *_T_*GNC-DOX after NIR laser irradiation shows greater mean fluorescence intensity compared to control, DOX alone, and *_T_*GNC-DOX without NIR laser treatment groups (*p* < 0.001, *p* < 0.01) (Figure 15B,C). This study reveals that *_T_*GNC-DOX can promote lipid peroxidation and ferroptosis following NIR treatment.

GNCs are biocompatible photothermal agent and widely explored in different diagnostic and therapeutic applications of cancer [38,39]. However, some recent reports reveal that PTT alone is insufficient for complete eradiation of tumors, and chances of cancer reoccurrence remain a probability [40]. Furthermore, PTT is not good choice for the metastatic tumors. Hence, combining chemotherapy with PTT makes a good approach for selectively killing cancer cells [41]. Additionally, incorporating chemotherapeutic agents in GNC can minimize systemic side effects of it by selective localization in tumor tissues [42].

The present investigation emphasizes on loading of DOX in *_T_*GNC through electrostatic interactions between positively charged DOX and negatively charged *_T_*GNC to form *_T_*GNC-DOX. While developing *_T_*GNC-DOX, the strategy was made to achieve a particle size of less than 100 nm to get enhanced permeability and retention effect for selective accumulation at tumor site. Moreover, the presence of HA helps in active targeting of *_T_*GNC-DOX towards overexpressed CD44 receptors in TNBC cells. The developed *_T_*GNC-DOX revealed high loading and entrapment efficiency. As well, the in vitro release profile showed pH and NIR-laser dependent DOX release in which higher amount of DOX release was observed at pH 6.5 in the presence of laser irradiation.

The cell-based assays such as cell viability, cellular uptake, intracellular ROS generation, apoptosis, and cell cycle analysis were performed to investigate the effectiveness of developed *_T_*GNC-DOX in MDA-MB-231 cells and results suggested good therapeutic potential of *_T_*GNC-DOX. A lipid peroxidation assay was performed to check whether developed *_T_*GNC-DOX had any role in ferroptosis and results revealed that *_T_*GNC-DOX possesses ferroptotic ability in MDA-MB-231 cells.

## 3. Materials and Methods

### 3.1. Materials

Gold (III) chloride trihydrate was purchased from Sigma Aldrich (St. Louis, MO, USA). Sodium hyaluronate (50 kDa) was acquired from Lifecore Biomedical Inc. (Chaska, MN, USA). Potassium bromide (KBr), sodium hydroxide (NaOH), and paraformaldehyde were procured from Sigma Aldrich (St. Louis, MO, USA). Sodium chloride and potassium chloride were obtained from Sigma Aldrich (Bangalore, India). Disodium hydrogen phosphate was procured from Thermo Fischer Scientific (Mumbai, India). Potassium dihydrogen phosphate was acquired from Merck (Mumbai, India). Leibovitz (L-15) medium, Fetal bovine serum (FBS), trypsin, and penicillin-streptomycin were obtained from Gibco (Waltham, MA, USA). 2′,7′-Dichlorofluorescein diacetate (DCFDA) was purchased from Abcam (Cambridge, UK). A dead cell apoptosis kit with annexin V was procured from Invitrogen (Waltham, MA, USA). Propidium iodide was purchased from Sigma Aldrich (St. Louis, MO, USA). BODIPY C11 probe was purchased from Invitrogen (Waltham, MA, USA). Human recombinant CD44 protein was purchased from Invitrogen (Mumbai, India). Mouse monoclonal anti-CD44 antibody and goat anti-mouse IgG H&L (Alexa Flour 647) were purchased from Abcam (Cambridge, UK).

### 3.2. Synthesis of _T_GNC-DOX

The *_T_*GNC-DOX was prepared from previously synthesized *_T_*GNC. Briefly, *_T_*GNC was synthesized by using HA as reducing agent in alkaline conditions at 150 °C using QbD approach and by varying the parameters such as gold chloride concentration, HA concentration and reaction time. The HA used in *_T_*GNC synthesis at as reducing agent for gold chloride, stabilizing agent and targeting ligand for CD44 receptor. These *_T_*GNCs were used to synthesize *_T_*GNC-DOX. Briefly, 0.4 mg of DOX was added in the 1 mL *_T_*GNC suspension and kept in an orbital shaker for 24 h at 37 °C temperature and 100 rpm.

### 3.3. Particle Size, Polydispersity Index, and Surface Charge Determination

Dynamic light scattering (DLS) phenomena of Malvern zeta sizer (Nano ZS, Malvern Instruments, Westborough, MA, USA) was used for the characterization of synthesized *_T_*GNC-DOX for polydispersity index (PDI), particle size, and surface charge. Disposable polystyrene cuvettes containing an aqueous dispersion of *_T_*GNC-DOX at 25 °C were used to measure particle size and PDI. Milli-Q water was used to dilute the sample for analysis of surface charge. The disposable folded capillary cell was used for surface charge analysis. All analyses were performed in triplicate (*n* = 3).

### 3.4. Surface Morphology

The surface morphology of *_T_*GNC-DOX was investigated by a field-emission scanning electron microscope (FE SEM, Gemini, Zeiss, Germany). The drop cast method was used to prepare the sample on a 1 cm^2^ glass slide, and then the slide was kept in a vacuum oven (OV-11, Jeio Tech, Columbia, DC, USA) for overnight drying and further analysis of the sample was done by FE SEM.

### 3.5. Topography Attributes

High-resolution transmission electron microscopy was used to evaluate the surface topography of *_T_*GNC-DOX (HR-TEM, Technai G2, F30FEI, Hillsboro, OR, USA). Firstly, suspension of *_T_*GNC-DOX was prepared, and then a copper grid was used to cast that sample. Finally, the sample was analyzed by HR TEM at an accelerating voltage of 300 kV.

### 3.6. UV Absorption Profile of _T_GNC-DOX

The absorption spectrum of *_T_*GNC-DOX was measured using the UV-Vis spectrophotometer (Evolution 300, Thermo Scientific, Waltham, MA, USA) by scanning over the wavelength of 400–850 nm. The Vision security software was used for UV spectrum analysis of *_T_*GNC-DOX.

### 3.7. Fourier-Transform Infrared (FTIR) Analysis

Fourier-transform infrared (FTIR) spectroscopy was used to identify the functional groups (Bruker alpha; Bruker, Ettlingen, Germany). DOX and *_T_*GNC-DOX were triturated with IR grade KBr, and thin pellets were obtained by pressing the mixture in a hydraulic press (6–8 tons pressure, Model M-15, techno search instruments, Mumbai, India). Final pellets were scanned in the range of 400 to 4000 cm^−1^ to obtain the FTIR spectra.

### 3.8. Confirmation of HA Surface Functionalization

The presence of HA in *_T_*GNC-DOX was confirmed by using confocal microscopy. In typical procedure, 1 mL each of HA functionalized DOX loaded *_T_*GNC-DOX and Tetrakis reduced conventional gold nanoparticles were taken and centrifuged at 5000 rpm for 10 min. The supernatant was discarded and obtained pellet dispersed in 100 μL recombinant CD44 protein (1 μg/mL). The dispersion was incubated for 30 min at room temperature to allow protein interact with nanoparticles. After 30 min, this dispersion was centrifuged again at 5000 rpm for 5 min, washed with PBS, centrifuged and pellet was dispersed in 100 μL anti-CD44 antibody (1 μg/mL). After 15 min incubation, dispersion was centrifuged at 5000 rpm for 5 min, washed with PBS, centrifuged and pellet was dispersed in 100 μL alexa flour 647 (0.5 μg/mL). After 15 min incubation, dispersion was centrifuged at 5000 rpm for 5 min, washed with PBS, centrifuged and pellet was dispersed in 50 μL PBS. Further, this pellet was mounted on glass slide and fluorescence intensity was measured using confocal laser scanning microscope (CLSM; Leica TCS SP5 AOBS, Leica, Wetzlar, Germany).

### 3.9. X-ray Diffraction Analysis

The crystalline nature of *_T_*GNC-DOX was verified by X-ray diffraction (XRD) using a powder X-ray diffractometer (Bruker D2 Phaser, Ettlingen, Germany). The diffraction peaks of DOX and TGNC-DOX were analyzed by keeping a 2 θ range between 5° and 80°. Different crystallinity of DOX and *_T_*GNC-DOX was analyzed from diffraction peaks.

### 3.10. Photothermal Response of _T_GNC-DOX

The photothermal response of *_T_*GNC-DOX to the NIR laser irradiation was determined for investigating the photothermal potential of *_T_*GNC-DOX. The 808 nm NIR laser was irradiated with the intensity of 2.4 w/cm^2^ for 15 min using 20 μg/mL *_T_*GNC-DOX suspension to check photothermal response. A thermal infrared camera was used to observe the temperature change (Optis PI, Berlin, Germany).

### 3.11. Reversible Photothermal Response

The impact of repeated NIR laser irradiation on the photothermal response of *_T_*GNC-DOX was evaluated to identify the reversible photothermal stability of *_T_*GNC-DOX. Initially, 808 nm NIR laser was irradiated for 15 min on 20 μg/mL suspension of *_T_*GNC-DOX at a laser power of 2.4 w/cm^2^, followed by a 15 min cooling period. This process was repeated, and a thermal infrared camera was used to continuously observe the temperature change.

### 3.12. Determination of Drug Loading and Entrapment Efficiency

For determination of drug loading and entrapment efficiency, 1 mL of *_T_*GNC-DOX suspension was centrifuged at 10,000 rpm for 10 min at 25 °C. Then the supernatant was collected, and the concentration of DOX in the supernatant was analyzed using a UV spectrophotometer at 479 nm. Drug loading and entrapment efficiency of DOX were measured using a UV spectrophotometer. The equations used to determine the loading efficiency and entrapment efficiency were as follows:(1)$$\mathrm{Loading}\;\mathrm{efficiency}\;\left(\%\right)=\frac{\mathrm{Wa}}{\mathrm{Wb}}\times 100$$
(2)$$\mathrm{Entrapment}\;\mathrm{efficiency}\;\left(\%\right)=\frac{\mathrm{Wa}}{\mathrm{Wt}}\times 100$$
where Wa is the weight of DOX present in *_T_*GNC-DOX, Wb is the weight of *_T_*GNC-DOX, Wt is the total weight of added DOX.

### 3.13. In Vitro Drug Release Studies

In vitro DOX release was performed under two conditions, i.e., at normal physiological condition (pH 7.4) and tumor microenvironment acidic condition (pH 6.5). In brief, 2 mL of *_T_*GNC-DOX suspension was filled in the dialysis bag (Molecular weight cut-off—10 kDa), and the dialysis bag shifted to a beaker containing 30 mL of phosphate buffer (pH 7.4) and acetate buffer (pH 6.5) medium. The temperature of the medium was kept at 37 ± 1.0 °C with constant stirring (100 rpm). Then, 1 mL of aliquot was removed from the medium at a predetermined time interval (0.5, 1, 2, 4, 6, 8, 12, 24, 48, 72 h) and replaced with a fresh medium. The amount of DOX released at different time points was analyzed at 479 nm using a UV spectrophotometer. For NIR laser/photothermal mediated drug release study, 2 mL of *_T_*GNC-DOX suspension was added to different six-well plates by maintaining pH conditions, i.e., 7.4 and 6.5. The temperature of the medium was kept at 37 ± 1.0 °C along with constant stirring (100 rpm). Then, *_T_*GNC-DOX suspension was irradiated with an 808 nm NIR laser for 15 min at a predetermined time interval (0.5, 1, 2, 4, 6, 8, 12, 24, 48, 72 h). After that, samples irradiated with NIR laser were centrifuged at 10,000 rpm for 10 min. The supernatant was collected and analyzed at 479 nm using a UV spectrophotometer.

### 3.14. Cellular Uptake Potential

Receptor-mediated endocytosis of *_T_*GNC-DOX was investigated by cellular uptake assay of *_T_*GNC-DOX in MDA-MB-231 cells. Initially, cells were seeded on a coverslip in a 6-well plate, and 24 h time was given to adhere the cells to the coverslip. HA (10 mg/mL) treatment was given for 2 h to the cells. Then after, *_T_*GNC-DOX treatment was given to the cells for 6 h and 24 h. After treatment exposure, cells were washed with the phosphate buffer saline (PBS), and 4% paraformaldehyde was used to fix the cells. Further, cell nucleus was stained using DAPI and cells were washed with PBS thrice. Then after, the slides were mounted using DPX mounting agent. Confocal laser scanning microscope (CLSM; Leica TCS SP5 AOBS, Leica, Wetzlar, Germany) and inductively coupled plasma atomic emission spectroscopy (ICP AES) (ARCOS, SPECTRO analytical instruments, Germany) was used to evaluate the cellular uptake of *_T_*GNC-DOX. For ICP AES, cells were seeded in the density of 5 × 10^5^ cells/well. After treatment exposure, cells were washed, trypsinized, digested, centrifuged, and gold concentration in the supernatant was evaluated using ICP-AES.

To study receptor-mediated endocytosis using anti-CD44 antibody, cells were seeded on coverslip in a 6-well plate, and 24 h time was given to adhere the cells. Anti-CD44 antibody (3 μg/mL) treatment was given for 1 h to the cells. Then after, *_T_*GNC-DOX treatment was given to the cells for 6 h. After treatment exposure, cells were washed with the phosphate buffer saline (PBS), and 4% paraformaldehyde was used to fix the cells. Further, cell nucleus was stained using DAPI and cells were washed with PBS thrice. Then after, the slides were mounted using DPX mounting agent. Confocal laser scanning microscope (CLSM; Leica TCS SP5 AOBS, Leica, Wetzlar, Germany) was used to evaluate the cellular uptake and receptor mediated endocytosis of *_T_*GNC-DOX.

### 3.15. Effect of Laser-Directed Thermal Ablation on Cytotoxicity _T_GNC-DOX in MDA-MB-231 Cells

The *IC*_50_ value of DOX was determined by varying the range of DOX from 0.1 µM to 1000 µM. Photothermal toxicity of *_T_*GNC-DOX was evaluated by Alamar blue assay using 808 nm NIR laser exposure to MDA-MB-231 cells. Initially, cells were seeded in a 96-well flat-bottomed culture plate at a density of 1 × 10^4^ cells/well, and 24 h was given to adhere the cells. Then, cells were incubated for 24 h after treatment of *_T_*GNC-DOX with a concentration of 20 µg/mL. After this, the 808 nm NIR laser with a power of 2.4 w/cm^2^ was irradiated on the *_T_*GNC-DOX treated cells for 15 min and incubated for 48 h. Alamar blue was added to each well of the plate to attain final concentration of 10 µg/mL. UV microplate reader (Multiscan GO, Thermo Fisher Scientific, Waltham, MA, USA) was used to measure the fluorescence at an excitation wavelength of 560 nm, and emission wavelength of 590 nm, and photothermal toxicity was evaluated.

### 3.16. Intracellular ROS Generation

2′,7′-Dichlorofluorescein diacetate (DCFDA) assay using flow cytometry was performed to determine the intracellular ROS generation. Cells were seeded in a 6-well plate at the density of 5 × 10^5^ cells/well and incubated for 24 h. Then, after giving *_T_*GNC-DOX treatment again, cells were incubated for 24 h. Further, 808 nm NIR laser with a power of 2.4 w/cm^2^ was irradiated on the *_T_*GNC-DOX treated cells for 15 min and incubated for 48 h. Finally, cells were washed with PBS, trypsinized, and then collected by harvesting them using centrifugation at 850 g. Cells were stained with DCFDA (10 µM) for 30 min to investigate the ROS generation by analyzing them using a flow cytometer (S3e™ Cell Sorter, Bio-Rad, New York, NY, USA).

### 3.17. Apoptosis Assay

Annexin-V FITC/PI kit was used to perform apoptosis assay. Cells were seeded in a 6-well plate at the density of 5 × 10^5^ cells/well and incubated for 24 h. Then, after giving *_T_*GNC-DOX treatment again, cells were incubated for 24 h. Further, 808 nm NIR laser with a power of 2.4 w/cm^2^ was irradiated on the *_T_*GNC-DOX treated cells for 15 min and incubated for 48 h. Then, cells were washed with PBS, trypsinized, and then collected by harvesting them using centrifugation at 850 g, and the supernatant was discarded. Cells were kept for 15 min after adding 50 µL 1× annexin binding buffer. Then, additional 200 µL 1× annexin binding buffer was added with the 2.5 µL of annexin-V/FITC and kept for 15 min. Finally, the cell suspension was analyzed using a flow cytometer after adding PI (1 µL, 1 mg/mL).

### 3.18. Cell Cycle Analysis

PI was used to perform a cell cycle assay. MDA-MB-231 cells were seeded in 6-well plates at a density of 5 × 10^5^ cells/well and incubated for 24 h. After that, *_T_*GNC-DOX treatment was given and further incubated for 24 h. Then, an 808 nm NIR laser with a power of 2.4 w/cm^2^ was irradiated on the *_T_*GNC-DOX treated cells for 15 min and incubated for 48 h. Then, cells were washed with PBS, trypsinized, and collected by harvesting them using centrifugation at 850 g. The cell fixation was performed by incubating cells with 70% ethanol for 30 min. Then cells were centrifuged to remove the ethanol. Further, 200 µL (50 µg/mL) of PI and 50 µL (100 µg/mL) of RNAse were added to cell suspension and kept for 15 min. Finally, the cell cycle of treatment groups was analyzed using a flow cytometer.

### 3.19. Lipid-ROS Detection

Lipid ROS generation was detected using a C11-BODIPY probe through the CSLM and flow cytometer. For the CSLM experiment, initially, 2 × 10^5^ MDAMB-231 cells/well were seeded on a coverslip in a 6-well plate, and 24 h time was given to adhere the cells to the coverslip. After that, cells were incubated for 24 h after treatment of *_T_*GNC-DOX with a concentration of 20 µg/mL. After this, an 808 nm NIR laser with a power: of 2.4 w/cm^2^ was irradiated on the *_T_*GNC-DOX treated cells for 15 min and incubated for 48 h. After 48 h treatment exposure, cells were washed with the phosphate buffer saline (PBS) and incubated with 2 µg/mL BODIPY C11 dye for 15 min. The coverslips were washed with PBS twice and analyzed using CSLM.

For flow cytometer experiments, cells were seeded in a 6-well plate at the density of 5 × 10^5^ cells/well and incubated for 24 h. Then after giving *_T_*GNC-DOX treatment again, cells were incubated for 24 h. Further 808 nm NIR laser with a power 2.4 w/cm^2^ was irradiated on the *_T_*GNC-DOX treated cells for 15 min and incubated for 48 h. Finally, cells were washed with PBS, trypsinized, and then collected by harvesting them using centrifugation at 850 g. Cells were stained with BODIPY C11 (2 µg/mL) for 30 min at 37 °C to investigate the lipid ROS generation by analyzing them by a flow cytometer.

### 3.20. Statistical Analysis

All the experiments were performed in triplicate. One-way ANOVA with Tukey’s multiple comparison test and Kolmogorov- Smirnov normality test was also used for statistical analysis using GraphPad Prism 6.01™ software (GraphPad Software Inc., San Diego, CA, USA). A probability level of *p* < 0.05, *p* < 0.01, and *p* < 0.001 was considered significant and highly significant. If values were found greater than 0.05, that would be considered non-significant.

## 4. Conclusions

Recently, chemo-PTT has shown great potential in cancer therapy. This investigation explores laser-responsive gold nanocomposite with cancer cell-specific CD44-receptor targeting efficiency (*_T_*GNC-DOX) to mediate chemo-photothermal ablation of MDA-MB-231 cancer cells. The developed *_T_*GNC-DOX was nanometric in size with high DOX-loading efficiency. The *_T_*GNC-DOX showed a potent rise in temperature following NIR 808 laser irradiation with good photothermal stability. *_T_*GNC-DOX possessed pH-responsive DOX-releasing properties. Under an acidic tumorous environment, *_T_*GNC-DOX showed an enhanced DOX-release profile compared to normal physiological pH of 7.4.

Interestingly, the release of DOX was further enhanced following NIR laser irradiation, indicating the thermo-responsive release behavior of DOX from *_T_*GNC-DOX. *_T_*GNC-DOX showed CD-44 receptor-specific uptake in MDA-MB-231 cells, as confirmed by confocal microscopy and ICP AES reports. The enhanced cellular uptake of *_T_*GNC-DOX followed by DOX cytosolic release and thermal ablation is corroborated with an enhanced cytotoxic effect following irradiation with an 808 nm NIR laser. The event of *_T_*GNC-DOX-directed photothermal ablation is accompanied by significant ROS generation and cellular apoptosis with maximum cell arrest in the G1 phase. Furthermore, the lipid peroxidation assay demonstrated that *_T_*GNC-DOX mediates high production of lipid-ROS, which was identified to be a prime reason for promoting ferroptosis in cancer cells. The NIR-808 laser-responsive thermal ablating and photothermal effect was found to be more effective than without NIR-808 laser-treatment approach, suggesting the fundamental role of photothermal ablation in cancer chemotherapy. The outcome concludes developed *_T_*GNC-DOX is a novel and potential tool to mediate laser-guided chemo-photothermal ablation treatment of cancer cells.

## Figures and Tables

**Figure 1 pharmaceutics-14-02734-f001:**
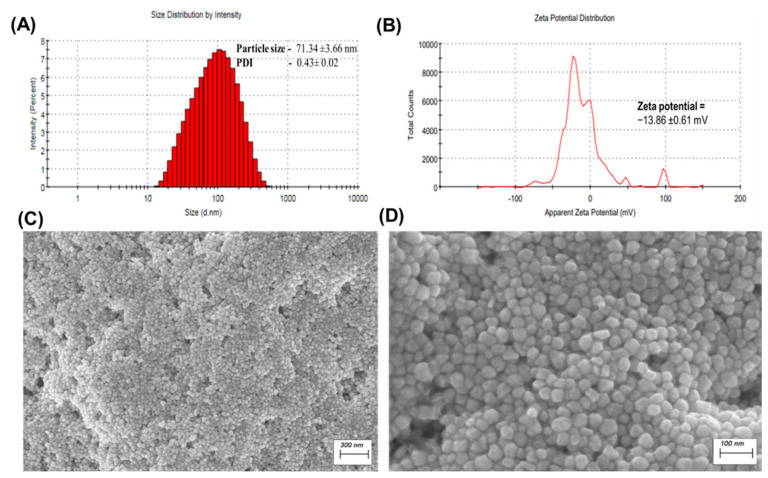
Characterization of *_T_*GNC-DOX. (**A**) Particle size histogram of prepared *_T_*GNC-DOX. (**B**) Zeta potential graph. (**C**) FE-SEM image of *_T_*GNC-DOX (scale: 300 nm). (**D**) Magnified FE-SEM image of *_T_*GNC-DOX (scale: 100 nm) (Represented values are *n* = 3).

**Figure 2 pharmaceutics-14-02734-f002:**
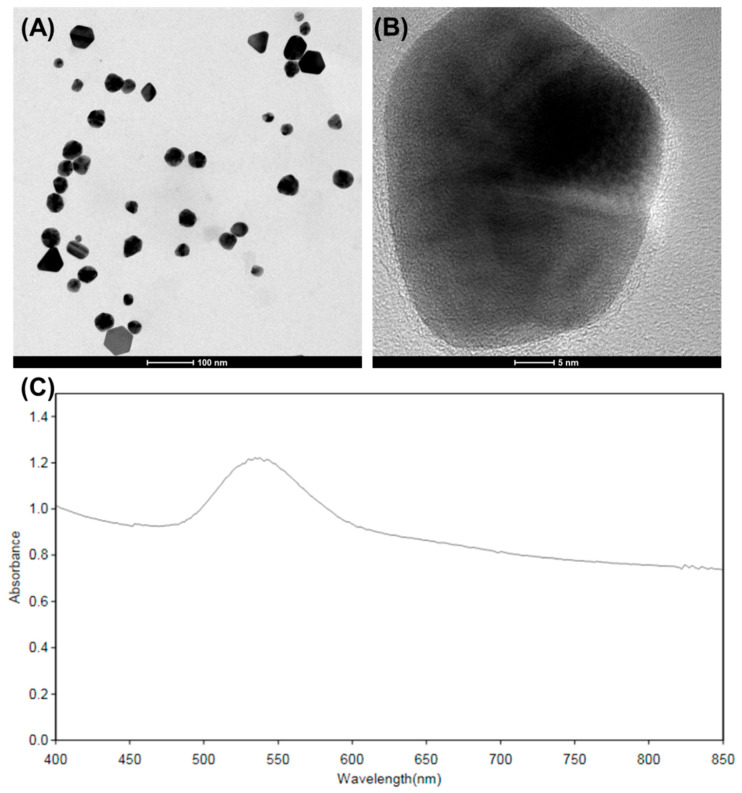
(**A**) TEM image of developed *_T_*GNC-DOX (scale: 100 nm). (**B**) Magnified TEM image of *_T_*GNC-DOX (scale: 5 nm). (**C**) UV absorption spectrum of *_T_*GNC-DOX.

**Figure 3 pharmaceutics-14-02734-f003:**
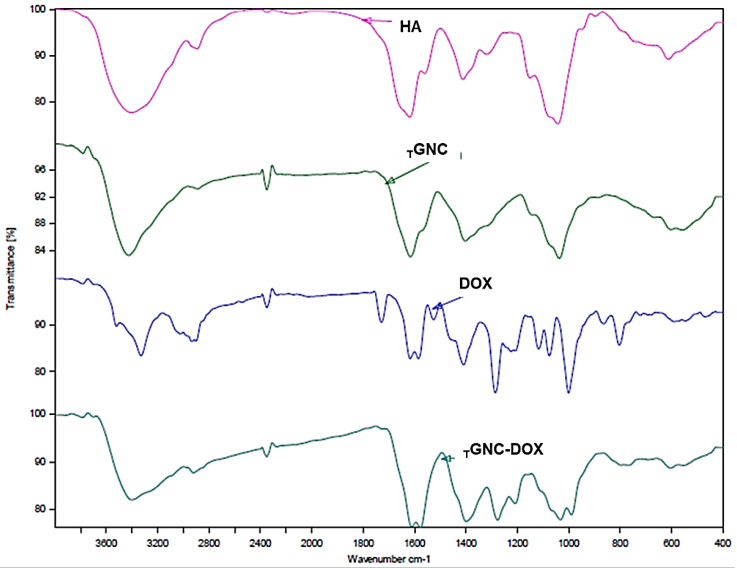
FTIR spectra of HA, *_T_*GNC, DOX, and *_T_*GNC-DOX.

**Figure 4 pharmaceutics-14-02734-f004:**
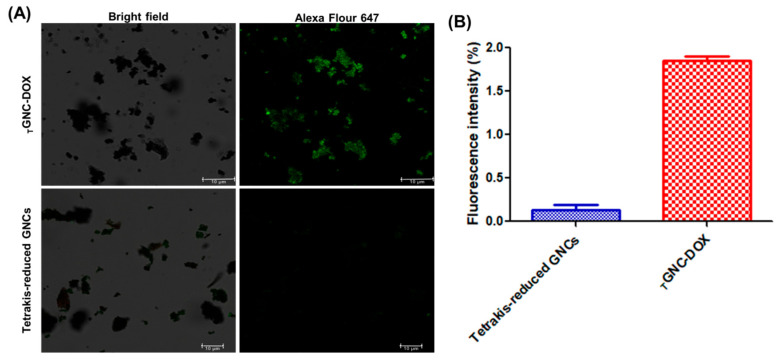
(**A**) The presence of HA in *_T_*GNC-DOX was confirmed by incubating *_T_*GNC-DOX and non-HA modified AuNPs i.e., Tetrakis-reduced gold nanoparticles with recombinant CD44 protein followed by anti-CD44 antibody treatment. Alexa fluor 647 was used as secondary antibody (green Fluorescence) (scale: 10 µm). (**B**) Bar graph showing fluorescence intensity of *_T_*GNC-DOX and Tetrakis-reduced gold nanoparticles.

**Figure 5 pharmaceutics-14-02734-f005:**
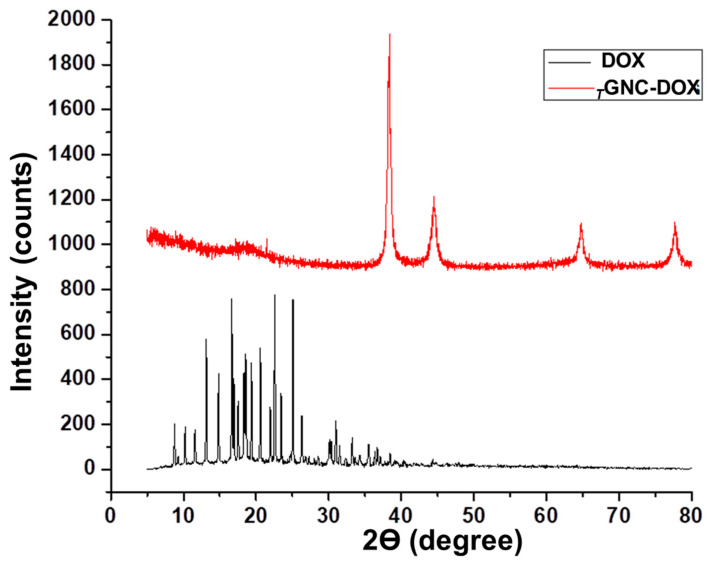
XRD pattern of DOX and *_T_*GNC-DOX.

**Figure 6 pharmaceutics-14-02734-f006:**
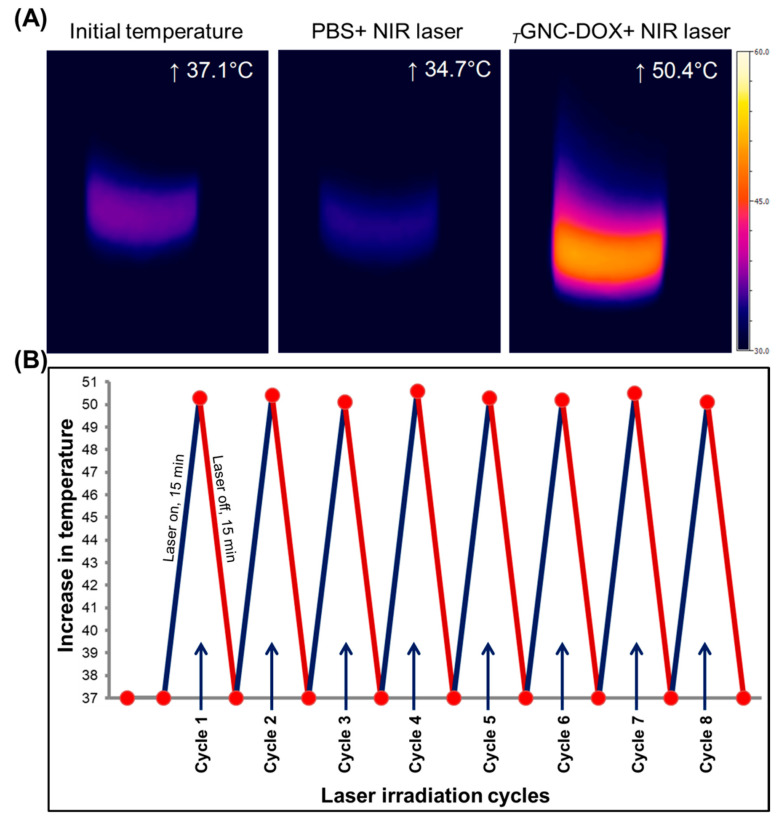
Photothermal response and photothermal stability of *_T_*GNC-DOX. (**A**) Increased temperature of different groups after 808 nm NIR laser exposure for 15 min at 2.4 W/cm^2^ recorded using NIR camera. (**B**) Photothermal stability of developed *_T_*GNC-DOX after 808 nm NIR laser irradiation is repeated.

**Figure 7 pharmaceutics-14-02734-f007:**
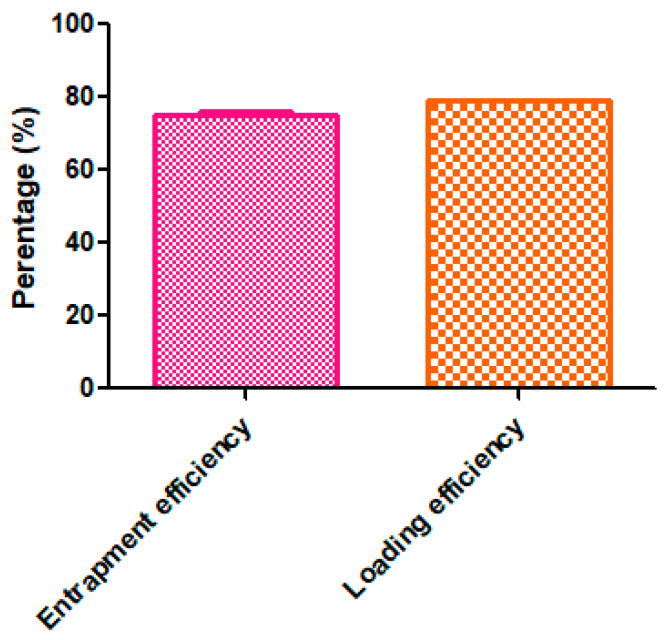
Percentage entrapment efficiency and loading efficiency of DOX in developed *_T_*GNC-DOX.

**Figure 8 pharmaceutics-14-02734-f008:**
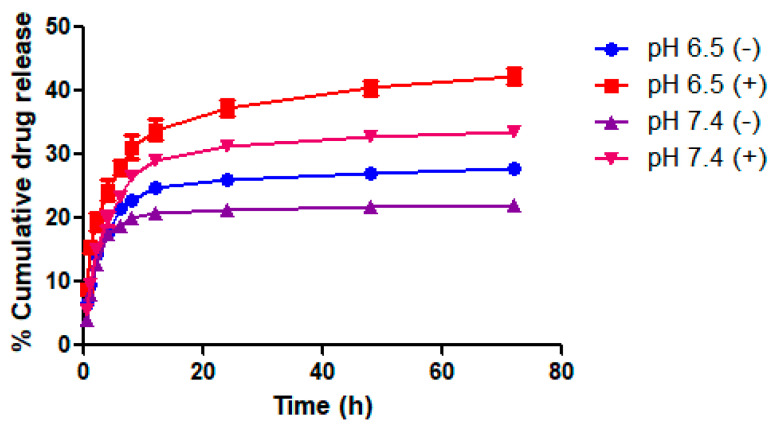
In vitro release behavior of DOX from *_T_*GNC-DOX at different pH conditions i.e., pH 6.5 and pH 7.4. The (−) indicates without NIR laser irradiation (+) indicates NIR laser treatment groups.

**Figure 9 pharmaceutics-14-02734-f009:**
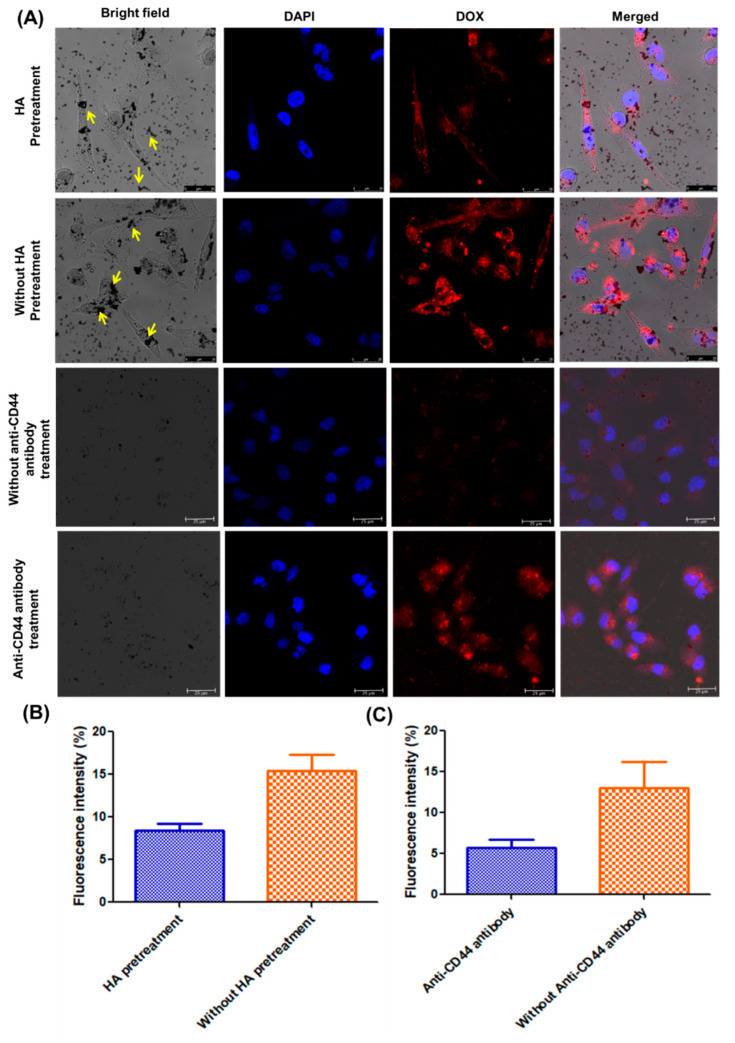
Cellular uptake study of *_T_*GNC-DOX in MDA-MB-231 cells. (**A**) CSLM images showing cellular uptake of *_T_*GNC-DOX followed by with or without HA pretreatment and after with and without blocking with anti-CD44 antibody (scale: 25 µm). (**B**) Bar graph showing percentage fluorescence intensity DOX with or without HA pretreatment. (**C**) Bar graph showing percentage fluorescence intensity DOX with or without anti-CD44 antibody pretreatment.

**Figure 10 pharmaceutics-14-02734-f010:**
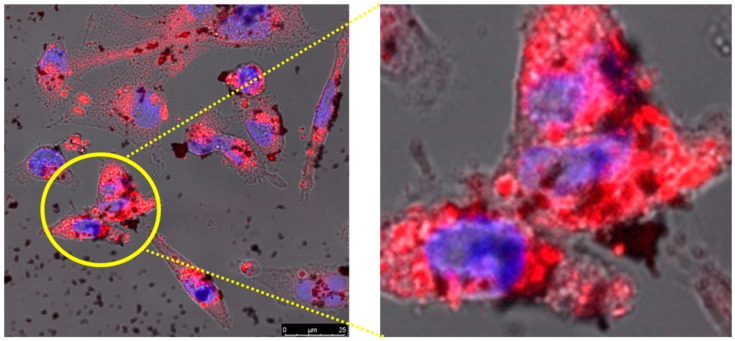
Intracellular release of DOX from *_T_*GNC-DOX. Red fluorescence indicates DOX (scale: 25 µm).

**Figure 11 pharmaceutics-14-02734-f011:**
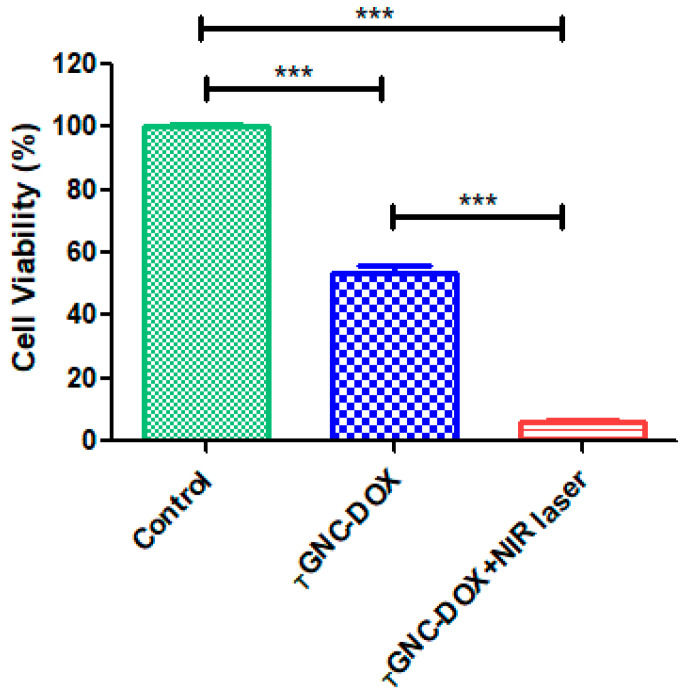
Cell viability assay of *_T_*GNC-DOX with and without NIR laser treatment (*** *p* < 0.001).

**Figure 12 pharmaceutics-14-02734-f012:**
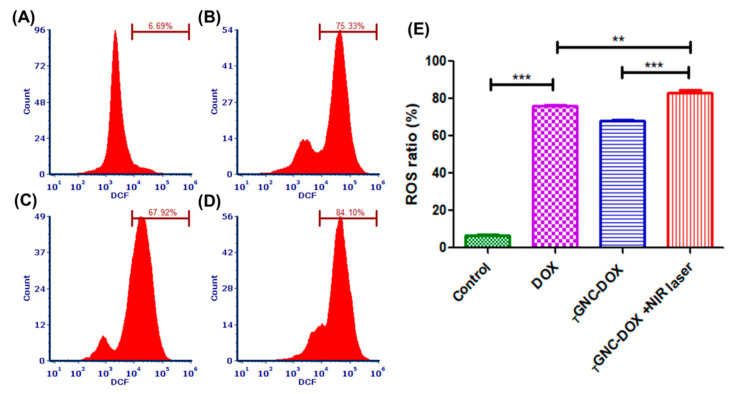
Intracellular ROS generation followed by treatment of: (**A**) control; (**B**) DOX; (**C**) *_T_*GNC-DOX only; (**D**) *_T_*GNC-DOX + NIR laser; and (**E**) bar graph of percentage ROS ratio of different treatment groups (*n* = 3) (*** *p* < 0.001, ** *p* < 0.01).

**Figure 13 pharmaceutics-14-02734-f013:**
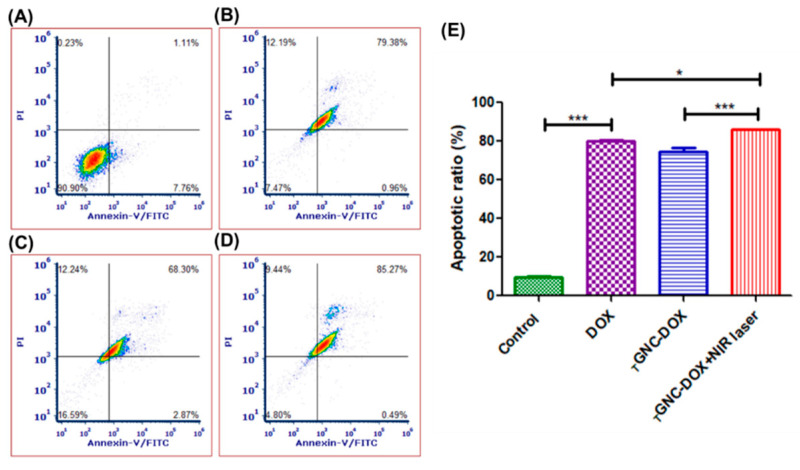
Apoptosis assay followed by treatment of: (**A**) control; (**B**) DOX; (**C**) *_T_*GNC-DOX only; (**D**) *_T_*GNC-DOX + NIR laser; and (**E**) bar graph of percentage apoptotic ratio of different treatment groups (*n* = 3) (*** *p* < 0.001, * *p* < 0.05).

**Figure 14 pharmaceutics-14-02734-f014:**
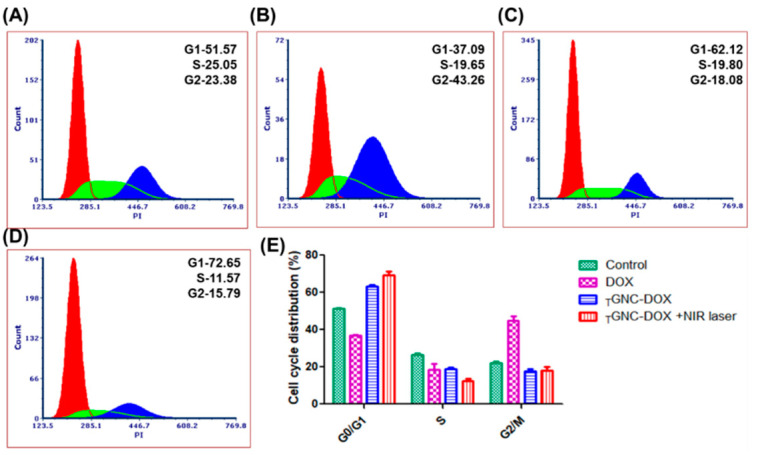
Cell cycle analysis followed by treatment of: (**A**) control; (**B**) DOX; (**C**) *_T_*GNC-DOX only; (**D**) *_T_*GNC-DOX + NIR laser; and (**E**) bar graph showing percentage cell cycle distribution of different treatment groups (*n* = 3).

**Figure 15 pharmaceutics-14-02734-f015:**
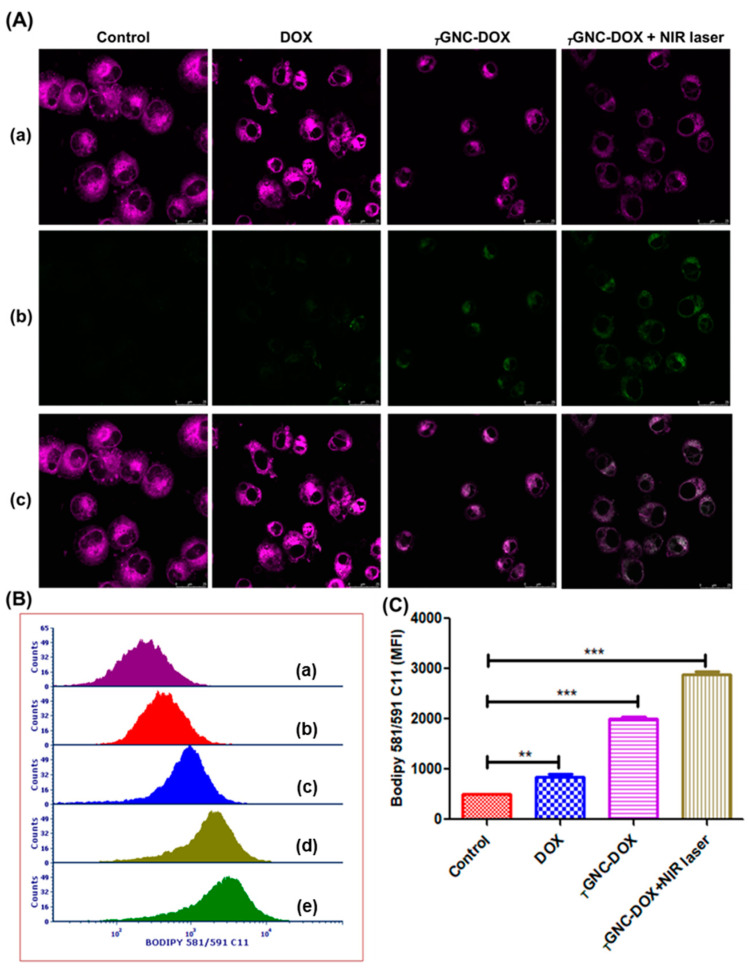
Lipid peroxidation level followed by treatment of: (**A**) CSLM images of different groups with (**a**) non-oxidized BODIPY C11 dye, (**b**) oxidized BODIPY C11 dye, (**c**) merged images. (**B**) Representative flow cytometry analysis fluorescent intensity images of different groups (**a**) unstained (**b**) control (**c**) DOX, (**d**) *_T_*GNC-DOX only, (**e**) *_T_*GNC-DOX + NIR laser. (**C**) Bar graph showing relative percentage mean fluorescence intensity of BODIPY C11 dye followed by different treatments (*n* = 3) (*** *p* < 0.001, ** *p* < 0.01) (scale: 25 µm).

## Data Availability

Not applicable.

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
