# Peer review of "CD44-Receptor Targeted Gold-Doxorubicin Nanocomposite for Pulsatile Chemo-Photothermal Therapy of Triple-Negative Breast Cancer Cells"

_pharmaceutics, 2022, doi:10.3390/pharmaceutics14122734_

Round 1

Reviewer 1 Report (Previous Reviewer 3)

It was a manuscript about the application of CD44 receptor-targeted gold-doxorubicin nanocomposite for chemo-photothermal therapy of triple-negative breast cancer cells. Here are some comments on this study that should be considered before publication:

1-      In my opinion, the first description that is mentioned at the beginning of the results is not suitable for there. Ä°t is better to transfer them after the results part.

2-      How did you confirm the presence of hyaluronic acid shell from TEM images?

3-      According to the results of SEM and TEM images what is the mean size of the nanoparticles?

4-      Please add the FTIR results of HA and TGNC alone for better confirmation.

5-      Figure caption of figure 9 needs to be rewritten. Please mention different parts of figure 9 in the figure caption.

6-      Section 3.6 need to be rewritten.

7-      “Then, TGNC-DOX suspension was irradiated with an 808 nm NIR laser at a predetermined time interval” for how long? 

Author Response

Response attached

Reviewer 2 Report (Previous Reviewer 2)

The author has answered the questions well. The Submission has been greatly improved, so I think it is acceptable for publication.

Author Response

Response attached

Reviewer 3 Report (Previous Reviewer 1)

All comments addressed. No more concerns.

Round 2

Reviewer 1 Report (Previous Reviewer 3)

-

This manuscript is a resubmission of an earlier submission. The following is a list of the peer review reports and author responses from that submission.

Round 1

Reviewer 1 Report

The manuscript reported the CD44 receptor-targeted gold-doxorubicin nanocomposite (TGNC-DOX) for pulsatile chemo-photothermal therapy of triple-negative breast cancer (TNBC). But the concept and design were lack of novelty and significance. Both gold nanoparticles and doxorubicin are widely applied for cancer therapy and the chemo-photothermal therapy strategy was widely used. In addition, hyaluronic acid has been widely used for tumor targeting. For these reasons I suggest to reject the paper for publication in Molecular. There are some points to be proved as follow:

1. The size and morphology of the particles are not uniform. How to ensure the physical and chemical properties of the materials?

2. Herein, negatively charged HA was modified on the surface of negatively charged TGNC-DOX. Please elaborate on the mechanism that how HA was modified to the nanoparticles?

3. More characterization should be provided to verify the successful modification of HA.

4. The laser power (2.4 W/cm2) was too high for photothermal therapy.

5. The DOX loading and entrapment efficiency was calculated to be 78.97 ±0.35% and 75.13 ±1.61%, respectively. Where is the data?

6. The author claimed that the CD44 receptor-targeted strategy was performed, but there is no relevant data to support this conclusion. The HA pretreatment can’t verify the CD44 receptor-mediated endocytosis of TGNC-DOX. The anti-CD44 antibody should be used.

7. The intracellular DOX release behavior should be studied.

8. The in vivo anti-tumor experiments were recommended.

9. How the significance data was obtained in the manuscript?

10. The scale bar should be provided in Figure 12A.

Reviewer 2 Report

The author prepared the CD44 receptor-targeted gold-doxorubicin nanocomposite  (TGNC-DOX) with laser-responsive and pH-dependent drug release behavior which could be used for pulsatile chemo-photothermal therapy and the effect was evaluated from cellular viability, cellular uptake, ROS generation, and apoptosis assays. However, the experimental design is not sufficient to support the conclusion. I suggest reconsidering this manuscript after addressing the following issue:

1.     The pH of the tumour microenvironment mentioned in the manuscript is 5.0, which seems to be different from the 6.5 reported in other literature, and the authors should verify the effect of drug release under this pH condition.

2.     For the introduction of “TGold nanoparticles (GNCs) the authors might wish to add a few more related references recently published in Chinese Chemical Letters.

3.     We noticed that the authors used a high intensity of NIR (2.4 W/cm2), which is higher than that reported in other literature. Is this intensity within the safe range? We are concerned that this light intensity may not be available for subsequent animal experiments.

4.     The authors have not calculated a specific value for the photothermal conversion efficiency, please provide the relevant data.

5.     Quantitative fluorescence data of Figure7 should be calculated.

6.     The scale bar in Figure 7 and Figure 12A should be more obvious

7.     The description of the experiment to verify the photothermal effect needs to be specific, e.g. in Figure5B, what is the exact duration of each cycle?

8.     The legend in Figure12B does not correspond to the images, e.g. a,b,c,d,e mentioned in the legend are not shown in Figure12B.

9.     Detailed synthesis steps of TGNC should be added in the experimental section.

Reviewer 3 Report

It was a manuscript about the application of DOX-loaded targeted nanoparticles for the aim of treating triple-negative breast cancer cells, here are some comments on this study that should be considered before publication:

1.     “To overcome this limitation of PTT, recently, combination strategies have been employed for the complete irradiation of tumors.” “irradiation” is not a good Word for here.

2.     “Gold nanoparticles (GNCs) have been widely used as a photothermal agent in PTT [8]. GNCs with different particle sizes and shapes have been employed for the PTT [9-11].” these are nearly the same sentences.

3.     There are several typos and grammatical mistakes in the text that should be corrected.

4.     The introduction part is poor. Please rewrite it.

5.     How did you separate drug-loaded nanoparticles from the free drug?

6.     “The obtained TGNC-DOX were characterized for particle size, zeta potential, surface morphology, crystalline nature, photothermal conversion efficiency, photothermal stability, entrapment efficiency and loading efficiency, etc.” this sentence should be deleted from the synthesis section.

7.     “X-ray diffraction (XRD) phenomena” phenomena is not a good word for here.

8.  “The photothermal conversion efficiency of TGNC-DOX was also determined for investigating the photothermal potential of TGNC-DOX.” please rewrite this sentence.

9.     “Wb is the weight of TGNC-DOX” “Wb” should be the weight of nanoparticles alone without any drug.

10.  It seems you have mistake in section 3.11, “Briefly, DOX solution (0.4 mg/mL) was added to 1 mL TGNC-DOX suspension and kept in an orbital shaker (100 rpm, 37 ±1.0 °C) for 24 h. Then, TGNC-DOX were centrifuged at 10000 rpm for 10 min at 25°C.” you were loaded DOX in the previous part, how did you load again?

11.  “tumor microenvironment acidic condition (pH 5.0).” pH 5 is not related to the microenvironment of the cancer cells, its pH is between 6.6-7.

12.  “HA (10mg/mL) treatment was given for 2 h to the cells.” why did you add hyaluronic acid to the cultured cells while you have hyaluronic acid in the structure of the nanoparticles?

13.  Lines 79-95 are not suitable for the first paragraph of the results section.

14.  “FE-SEM studied surface morphology”, this is not true, please rewrite it.

15.  “FE-SEM images of GNC-DOX showed different shapes like a rod, spherical, hexagonal, triangular, etc.,” why do nanoparticles have different shapes?

16.  “The UV spectrum showed an increase in the absorption band in the NIR 136 region” this couldn’t be seen in figure 2C.

17.  Please add the UV-Visible spectrum of GNC, GNC-HA, and GNC-DOX for better comparison.

18.  “FTIR peaks at 1577 cm−1 and 1611 cm−1 confirm C=C ring stretching; peaks at 797 cm−1 and 692 cm−1 depict the C=H bend and C=C ring bend, respectively” these peaks related to which component of the nanoparticle? Please add the FTIR spectrum of GNC-HA. The same for XRD analysis.

19.  Why do you choose 15 min as the irradiation time?

20.  What was the ratio between the amounts of nanoparticles and DOX?

21.  “cellular uptake of TGNC-DOX was found to be higher, i.e., 5.07 ±0.25 µg intracellular gold concentration, compared to pretreatment with HA, i.e., 4.72 ±0.16 µg after 6 h” according to these results there is no significant differences between the amounts of cellular uptake in both samples. How do you explain it?

22.  “Furthermore, CSLM results revealed that TGNC-DOX, after entering into cells, causes the release of DOX” from where do you reach to this result?

23.  You need to have a better discussion.